# Key Technologies for an Orchard Variable-Rate Sprayer: Current Status and Future Prospects

**Zhiming Wei** [1,2], **Xinyu Xue** [2,*], **Ramón Salcedo** [3], **Zhihong Zhang** [4], **Emilio Gil** [3], **Yitian Sun** [1], **Qinglong Li** [1], **Jingxin Shen** [1], **Qinghai He** [1], **Qingqing Dou** [1] and **Yungan Zhang** [5]

1. Shandong Academy of Agricultural Machinery Sciences, Jinan 250100, China
2. Nanjing Institute of Agricultural Mechanization, Ministry of Agriculture and Rural Affairs, Nanjing 210014, China
3. Department of Agro-Food Engineering and Biotechnology, Polytechnical University of Catalonia, 08034 Castelldefels, Spain
4. Faculty of Modern Agricultural Engineering, Kunming University of Science and Technology, Kunming 650500, China
5. School of Mechanical Engineering, University of Jinan, Jinan 250024, China
* Correspondence: xuexinyu@caas.cn

**Abstract:** An orchard variable-rate sprayer applies the appropriate amount of plant protection products only where they are needed based on detection data from advanced sensors, a system that has attracted increasing attention. The latest developments in the detection unit, variable control unit, and signal-processing algorithm of the variable-rate sprayer are discussed. The detection of target position and volume is realized with an ultrasonic sensor, a laser scanning sensor, or other methods. The technology of real-time acquisition of foliage density, plant diseases and pests and their severity, as well as meteorological parameters needs further improvements. Among the three variable-flow-rate control units, pulse width modulation was the most widely used, followed by pressure-based, and variable concentration, which is preliminarily verified in the laboratory. The variable air supply control unit is tested both in the laboratory and in field experiments. The tree-row-volume model, the leaf-wall-area model, and the continuous application mode are widely used algorithms. Advanced research on a variable-rate sprayer is analyzed and future prospects are pointed out. A laser-based variable-rate intelligent sprayer equipped with pulse width modulation solenoid valves to tune spray outputs in real time based on target structures may have the potential to be successfully adopted by growers on a large scale in the foreseeable future. It will be a future research direction to develop an intelligent multi-sensor-fusion variable-rate sprayer based on target crop characteristics, plant diseases and pests and their severity, as well as meteorological conditions while achieving multi-variable control.

**Keywords:** orchard variable-rate sprayer; target detection; ultrasonic sensor; LiDAR sensor; pulse width modulation; intelligent sprayer

## 1. Introduction

The application of pesticides in orchards and nurseries ensures the quality and yield of fruits and plants. However, the current constant-rate spray system applies pesticides uniformly over orchards and nurseries, failing to take into account the spatial variations of crop characteristics and severity of diseases and pests throughout these places [1–3]. It continuously discharges excessive pesticides into off-target and sparse areas [4], resulting in agrochemical waste, production cost increment, and environmental contamination [5–9].

With the growing awareness of environmental protection and the great demand for healthy fruit, more efficient intelligent spray systems and methods are in urgent need to cut down on plant protection products (PPPs) while ensuring spray efficacy. One promising strategy to optimize the application is to adjust spray outputs according to real-time

detection data obtained with sensors. This sensor-based variable-rate spray system applies the appropriate amount of pesticides only to the target based on its structural characteristics. Therefore, it significantly lowers agrochemical use and environmental pollution, and maximises spray efficacy, biological effect, and profit margin, thereby achieving an environmentally friendly agriculture production [10,11].

Generally, a variable-rate spray system is composed of a detection unit, a data processing algorithm, and a variable control unit. To be specific, the detection unit uses advanced sensors to detect target parameters, plant diseases and insect pests and their severity, and the meteorological conditions of the target growth area. The data processing algorithm, stored in the computer, first translates the detection information from the detection unit into an expected flow rate or other operating parameters of the sprayer and then converts the intended flow rate or air volume into the control signal of the corresponding actuator. The variable control unit carries out the variable-flow-rate spray output, the variable concentration, the variable air supply, or the variable nozzle position and type based on the control signal.

The variable-rate spray system has become a research focus in the past few decades and major advances have been reached, but there are still technical difficulties to overcome. In order to promote research on the orchard variable-rate spray system, this review summarizes the research status of the key technologies of three detection units (Target parameter, Plant diseases and pests and their severity, and Meteorological conditions) and four variable control units (Variable flow rate, Variable concentration, Variable air supply, and Variable nozzle position and type) and points out its future development directions.

This review is organized as follows: the detection unit, variable control unit, and signal processing algorithm will be discussed in Sections 2–4, respectively. Section 5 will comprehensively summarize the research status of the variable-rate sprayer. Future development directions will be pointed out in Section 6.

## 2. Detection unit

### 2.1. Target Parameter Detection

Target parameters that needed detecting were target position, volume, and foliage density. Real-time acquisition of target parameters was the first important step in developing a variable-rate spray system. For instance, Rosell and Sanz [12] analyzed in detail various sensors used to detect target parameters, including radar systems, X-ray, nuclear magnetic resonance imaging (NMRI), digital imaging cameras, light sensors, stereovision, ultrasonic sensors, and light detection and ranging (LiDAR) sensors. Through comparing their advantages and disadvantages, they noted that LiDAR was the most promising detection sensor because it could be used to obtain a high-precision three-dimensional model of the target, although the ultrasonic sensor was still an attractive choice because of its low cost. The physical principles, advantages, and disadvantages of the ultrasonic sensor and the LiDAR sensor are listed in Table 1.

**Table 1.** Physical principles, advantages, and disadvantages of the ultrasonic sensor and the LiDAR sensor [13].

| Sensor | Measuring Principle | Advantage | Disadvantage |
|---|---|---|---|
| Ultrasonic sensor | Measure the flight time period from the time when an ultrasonic wave is emitted to the time when it hits the target and reflects back to determine the distance from the sensor to the target | 1) Simplicity<br>2) Low cost<br>3) Relatively high robustness | 1) Relatively wide divergence angle<br>2) Low spatial resolution<br>3) Interference between adjacent sensors |

**Table 1.** *Cont.*

| Sensor | Measuring Principle | Advantage | Disadvantage |
|---|---|---|---|
| LiDAR sensor | 1) Based on the time-of-flight principle to meter the emitted laser travel time between the sensor and the target<br>2) Based on the phase-shift principle to measure the phase difference between the incident and reflected laser beams | 1) More accurate than ultrasonic sensor<br>2) Suitable for a variety of environmental conditions | 1) Expensive<br>2) Easy to produce spatial position errors for 2D LiDAR compared with 3D LiDAR |

2.1.1. Ultrasonic Sensor

The ultrasonic sensor, consisting of a wave emitter, a chronometer, and a wave receiver, was typically used in the variable-rate sprayer before the advent of the laser scanning sensor, with its measuring principle shown in Table 1 [14–20]. Based on the measured distance, the canopy width, height, area, and volume could be estimated. Its principle of measuring the target volume was as follows. Several ultrasonic sensors were installed at different heights, horizontally cutting the target into corresponding independent units. Assuming that the distance from the sensor to the target centerline was constant and the target was symmetrical, the thickness of each unit was calculated based on the measured distance from the sensor to the leaf periphery. Together with the known travel speed, the target volume was estimated (Figure 1a). For example, Tumbo et al. [21] used 20 ultrasonic sensors (ten per side) to detect the target volume in real time. Gil et al. [14] incorporated three ultrasonic sensors per side and three solenoid valves into an air-assisted sprayer. These sensors were normally mounted at different heights of the sprayer, corresponding to the upper, middle, and lower parts of the target. Each sensor was responsible for the target presence detection and the target volume measurement in the corresponding area.

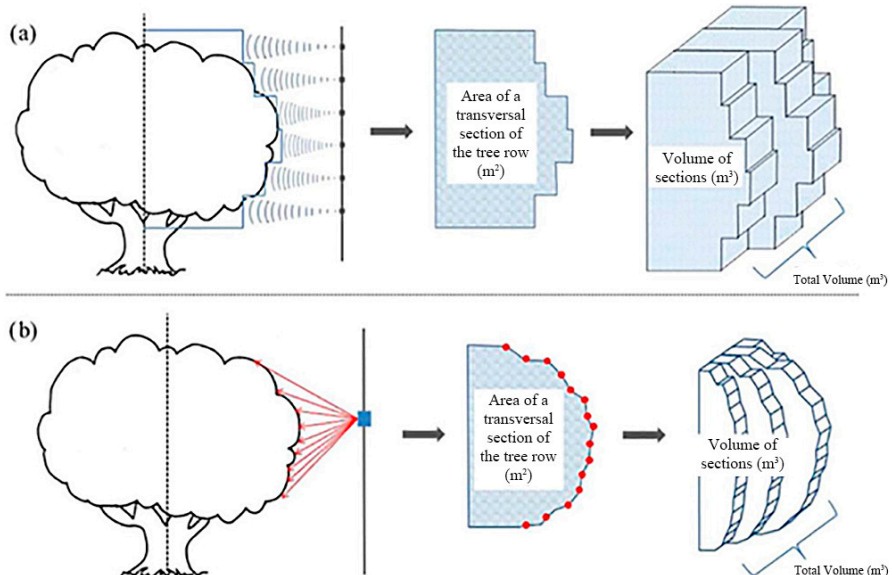

**Figure 1.** Measurement of target volume with ultrasonic (**a**) and LiDAR (**b**) sensors [22].

The ultrasonic sensor could also be used to assess the canopy density based on the principle that the ultrasound echoes were proportional to the canopy density; namely, the denser the canopy was, the more ultrasound echoes there would be (Figure 2) [16,23]. Its main advantages were simplicity, low cost, and relatively high robustness during harsh field conditions [24,25]. However, its accuracy was affected by many factors such as detection

distance, travel speed, as well as the temperature and humidity of the transmission medium. When the sensor-to-target distance increased, it had a relatively wide divergence angle and low spatial resolution [12]. When multiple ultrasonic sensors operated simultaneously, there was interference between adjacent sensors [21,26]. For illustration, Palleja et al. [27] proved that uneven ground would cause an error in measuring the distance from the sensor to the tree, and that a small margin of error would lead to a large deviation in the volume estimation (with errors up to 30% for an error of 50 mm). Jeon et al. [28] investigated the ultrasonic sensor's durability and accuracy for detecting canopy under simulated environmental and operating conditions such as cold temperatures, dust clouds, travel speeds, crosswinds, and spray clouds. Their test results showed that the ultrasonic sensor had the capability to detect the canopy presence, size, volume, and density with acceptable precision and could be used in the variable-rate spray system.

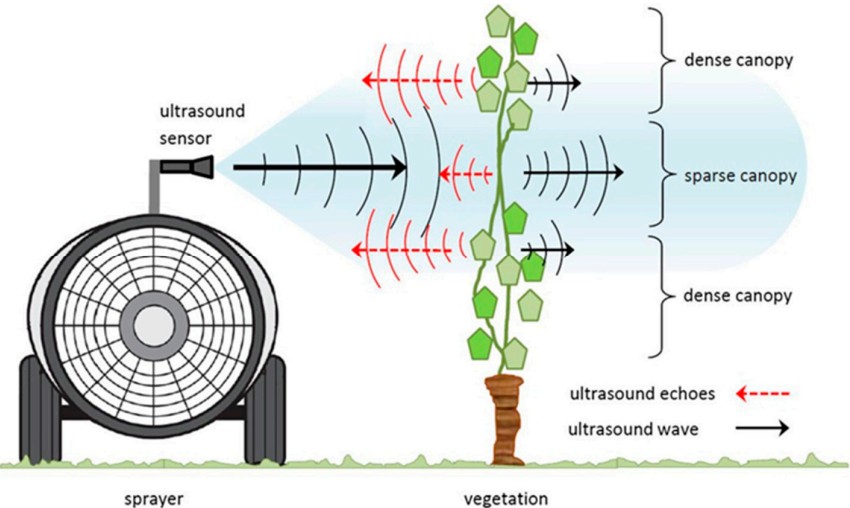

**Figure 2.** Schematic diagram of canopy density measurement with the ultrasonic sensor [16].

### 2.1.2. LiDAR

At present, LiDAR is the most widely used non-destructive remote sensing technology [29]. It was based on two methods to meter the distance from the sensor to the target (Table 1) [12]. Generally, the canopy width, height, and volume were detected using the time-of-flight (TOF) principle. Its working principle to estimate the canopy volume was as follows. The laser beam emitted from the sensor intersected with the canopy to form many measurement points according to its resolution. The distance and the relative angle from each measurement point to the sensor were presented in polar coordinates. The canopy thickness corresponding to each measurement point was calculated through coordinate transformation. The canopy volume could be estimated with the calculated canopy thickness, the measured canopy height, and the known forward speed (Figure 1b). Compared with the ultrasonic sensor, the LiDAR sensor was more expensive. However, it could more quickly and accurately measure target characteristics in varying environmental conditions except the dusty environment [21,22,30–34].

The laser scanning sensor, one kind of LiDAR sensor, was considerably investigated and applied [30,31,35–38]. This sensor had a high-precision step motor rotating the laser beam regularly to detect the target surface. Chen et al. [24] first successfully integrated a laser sensor with a 180° radial range into an experimental variable-rate atomizer to discriminate tree-geometric features and individually adjust the spray discharge of 20 nozzles in real time. However, it could only recognize targets on one side of the atomizer due to its radial-range limitation. To detect targets on both sides of the atomizer, two of these sensors were required, leading to higher costs and more difficulties in synchronization of automatic control functions. Later, Liu et al. [39] introduced a wide radial-range laser sensor to replace two 180° radial-range sensors, successfully resolving these problems. This

new sensor could release 1080 laser beams during each scanning cycle (25 ms) to detect trees on both sides of the atomizer in a 270° radial range with an angular resolution of 0.25°. Liu and Zhu [40] evaluated its accuracy in measuring target sizes with complex shapes under laboratory conditions, with their test results showing that its accuracy did not vary with the target's shape, size, and detection distance.

Due to cost factors, widely-used laser scanning sensors in variable sprayers were two-dimensional. The three-dimensional data of the target were obtained by the detection system moving along the target row with a constant speed. However, this could easily produce spatial position errors due to the influence of system trajectories, travel speeds, ground conditions, and the external environment [41]. To resolve these problems, Qiao et al. [42] developed a variable-rate spray system with a 16-line 3D LiDAR sensor. This sensor could collect the three-dimensional target data in a real-time and direct way. It was shown by their test results that its maximum error, minimum error, and average error were 8.42%, 0.17%, and 4.59%, respectively, and that it had good accuracy in scanning and identifying the crop height.

### 2.2. Detection of Plant Diseases and Pests and Their Severity

The accurate detection of plant diseases and insect pests along with their severity was a first step in developing an on-demand variable spray system. There were direct and indirect methods to diagnose plant diseases and insect pests. The direct one was dependent on serological technology [43] or molecular technology [44] to detect plant diseases and insect pests under laboratory conditions. This method enjoyed mature technology and high accuracy, but the detection process was relatively complex and time-consuming, so it was inappropriate to online real-time detection [45].

The indirect method was mainly based on the plant morphological changes and volatile organic compound (VOC) profiles to determine the occurrence of plant diseases and pests. Machine vision technology was responsible for detecting plant morphological changes [46], while the electronic nose (E-nose) was responsible for detecting the VOCs.

### 2.2.1. Machine Vision

Machine vision technology employed equipment and algorithms to obtain and process images used to determine whether there were plant diseases and pests in the collected plant images [46]. The detection process for plant diseases and insect pests based on machine vision technology mainly consisted of image acquisition, image preprocessing, image segmentation, and image classification. The specific process was as follows. First, a digital camera was used to acquire raw images from the field. Then, the raw images were processed to obtain the expected images by performing image preprocessing techniques such as gray transformation, filtering, and resizing. Next, the images were segmented using different segmentation algorithms. Finally, the segmented images were input into the classifier for image classification, thus obtaining the recognition results of plant disease and pest images.

Singh and Misra [47] performed image segmentation using a genetic algorithm, and found that the proposed algorithm had the capability and efficiency for plant leaf disease recognition. Huang et al. [48] detected Helminthosporium Leaf Blotch (HLB) disease based on an unmanned aerial vehicle (UAV) for image acquisition and the convolutional neural network (CNN) for disease classification. Their test results indicated that UAV-based machine vision technology could precisely discriminate between healthy and infected areas with the overall accuracy and standard error of the CNN method reaching 91.43% and 0.83%, respectively. Chen et al. [49] proposed the enhanced artificial neural network (EANN) for image segmentation and the convolutional neural network (CNN) for image classification so as to recognize plant diseases. Their test results demonstrated that compared with the conventional approach, the proposed combined algorithms presented a higher accuracy and efficiency (Figure 3).

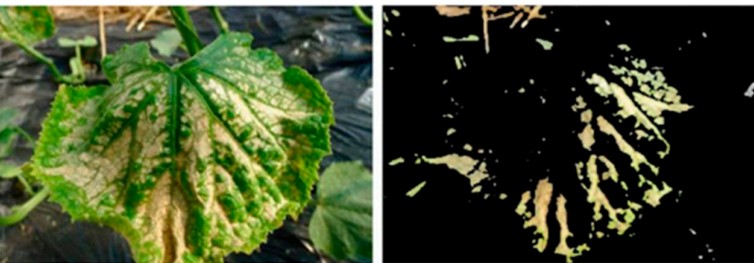

**Figure 3.** Input image and detection result for plant disease diagnosis with image technology. The leaf has lost its color due to disease attack [49].

Although the achievement in machine vision studies was amazing, and the detection accuracy and efficiency improved greatly, there was still a long way to go for the real-time detection of diseases and pests in the field based on mobile devices. The following problems were frequently encountered in real-time detection: small dataset size; relatively slow detection speed; and external disturbances such as nonuniform illumination, dense occlusion, blurred equipment dithering, etc. These challenges made it difficult to apply machine vision directly to the mobile devices of variable sprayers [46].

2.2.2. E-Nose

The E-nose was designed based on the principle that when plants are attacked by diseases and pests, plants will release specific VOCs in real time to fight the attackers. It mimicked the human olfactory system and diagnosed plant diseases and insect pests and their severity by detecting the distribution of volatile gases. The E-nose consisted of gas sensors, signal conditioning circuits, and pattern recognition algorithms. When the gas sensors detected volatile gases, the sensing material of sensors changed, causing electrical property changes. Based on this principle, gas sensors transformed the VOC information into electronic signals. Then signal conditioning circuits modulated these signals by de-noising, amplifying, and AD conversion. Finally, pattern-recognition algorithms classified the gas and recognized the diseases and pests (Figure 4).

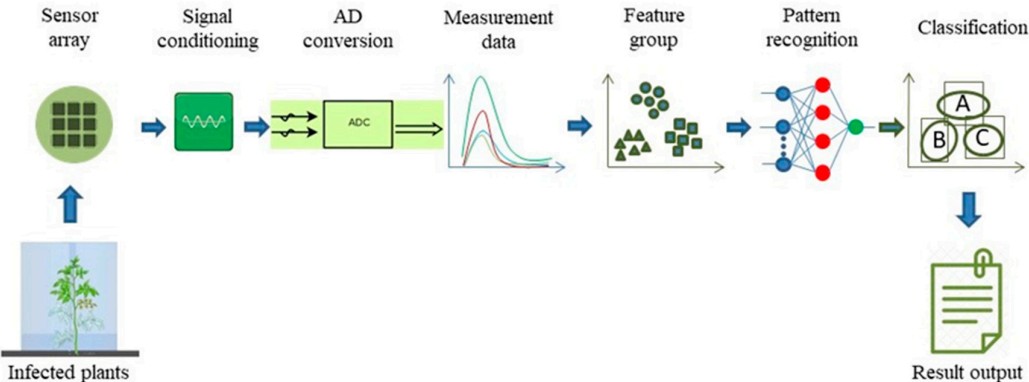

**Figure 4.** Process chart for diagnosing plant diseases and pests using the electronic nose.

For example, Soh et al. [50] developed an E-nose to classify aromatic herbs. Two classification algorithms used in this E-nose were artificial neural network (ANN) and adaptive neuro-fuzzy inference system (ANFIS). It was indicated that the E-nose with ANFIS presented 94.8% accuracy and could successfully classify the species of the aromatic herbs. Sun et al. [51] used an E-nose for early detection of *Botrytis cinerea* infestation of the tomato plant and demonstrated that this early diagnosis with the E-nose was feasible and possible. Oates et al. [52] applied an E-nose to detect lethal bronzing disease in cabbage palms, and discovered that the E-nose could discriminate healthy leaves from diseased leaves, but could not detect the severity of infection. Şennik et al. [53] developed an E-nose

system based on a functionalized capacitive micromachined ultrasonic transducer (CMUT) array for selective detection of plant volatiles. This system used the k-nearest neighbor (k-NN) algorithm for gas classification, suggesting that the sensor array and the k-NN classifier could selectively diagnose plant diseases with a minimum of 98% accuracy, even at different relative humidity levels.

Under laboratory conditions, the E-nose proved to be a promising technology for the rapid and early detection of plant diseases and pests. Its advantages were operational simplicity, non-destructivity, and bulk sampling [54]. However, in the field application, many challenges needed overcoming. The temperature and humidity of the field were uncontrollable and kept changing, which influenced the sensor's accuracy and lifetime [45]. Furthermore, the gas compositions of the open field were complex, and the concentration of VOCs released by plants was comparatively low, so the background noise could easily cover up the changes in real plant VOCs. Hence, E-nose technology needed to be further investigated and improved, especially for real-time applications in the field.

### 2.3. Detection of Meteorological Conditions

Meteorological conditions to be detected were wind direction, wind speed, temperature, relative humidity, barometric pressure, cloud cover, and rain probability at the application site. There was corresponding equipment and sensors for their detection. Considerable research studied the effect of meteorological conditions on spray performance [55–58]. For instance, Nuyttens et al. [56] detected wind speed, wind direction, temperature, and relative humidity using a Campbell scientific weather station supporting sensor in the test field and studied the effect of these weather factors on spray drift from field sprayers. Arvidsson et al. [57] measured air temperature and relative humidity with a Rotronic YA-100 hygrometer, wind speed and direction with an omnidirectional thermal anemometer and a wind vane, and barometric pressure with a meteorological station close to the experimental site, and graded the cloud cover by visual observation. It was indicated by their field trials that among all the environmental factors, the most decisive factors influencing the spray drift would be wind speed, followed by air temperature. Bahrouni et al. [58] measured temperature, relative humidity, and wind speed using a multifunction measuring instrument, and assessed the impact of wind speed on droplet foliage deposition and soil losses in the field. The objectives of most of this research were to investigate the impact of the meteorological conditions on spray performance, to provide data to spray operators, and to let them make sound judgments and adjust the operating parameters of sprayers based on the average values of meteorological variables before spraying pesticides.

Nevertheless, the local meteorological conditions varied greatly during the pesticide application process. It was obviously defective to adjust the operating parameters of sprayers based on the average values of the meteorological variables before spraying. Therefore, it became pressing to design a sprayer capable of real-time adjustment of the operating parameters based on the instantaneous meteorological conditions. However, this variable sprayer was scarce. The operational parameters to be adjusted included spray timing, nozzle type, and air supply volume. Balsari et al. [15] put forward the concept of a variable orchard sprayer that could automatically adjust the spray application parameters according to the environmental conditions at the time of spraying. Hołownicki et al. [59] designed a variable air-assisted (VAA) sprayer with two independently operating fans producing two air jets blowing in opposite directions. A detection sensor was used to detect the natural wind speed in real time during the application. This sprayer could adjust the air volume from each fan and orient the air direction of each air jet based on the meteorological information monitored. The authors acknowledged that the sprayer was still a concept in terms of regulating air volume and air direction according to natural wind speed and direction. Zhai et al. [9] proposed that the speed and direction of the wind should be considered when calculating the air volume and air speed demand of an air-assisted sprayer.

Online real-time detection of orchard targets was the premise and foundation of developing variable-rate sprayers. Target information detection methods and their corresponding development levels of variable-rate sprayers are displayed in Table 2.

**Table 2.** Target information detection methods and their corresponding development levels in variable-rate sprayers.

| Target Detection | Sensor | Development Level |
|---|---|---|
| Target position and volume | Ultrasonic sensor; LiDAR sensor | Mature technology and commercialized application |
| Canopy density | Ultrasonic sensor; LiDAR sensor | Preliminarily verified in laboratory and field experiments |
| Plant diseases and pests and their severity | Machine vision; E-nose | In the stage of laboratory research |
| Meteorological conditions | Meteorological sensor | In the "conception" stage |

## 3. Variable Control Unit

### 3.1. Variable Flow Rate Control Unit

Figure 5 is a schematic diagram of the variable-rate spray system. This system needed to mix pesticide concentrates with water in the pesticide tank before spraying. It achieved variable flow rate applications using three approaches: (1) adjusting the operating pressure; (2) adjusting the duty cycle of the PWM solenoid valve; and (3) changing nozzle orifice size and shape. It should be pointed out that Figure 5 is the integration of the three approaches, with only one of them required to achieve the variable flow rate in a variable-rate spraying system.

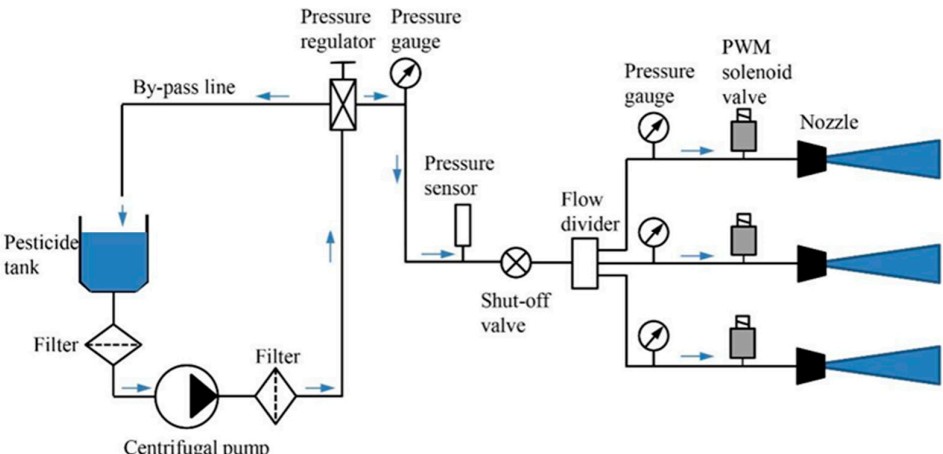

**Figure 5.** Schematic diagram of the variable rate spraying system.

#### 3.1.1. Pressure-Based Flow Rate Control Unit

The principle of the pressure-based flow rate control unit was to adjust the operating pressure of the spray system to achieve variable flow rate applications. There were two main means to change the operating pressure: (1) adjusting the opening of the pressure regulator; and (2) adjusting the rotation speed of the hydraulic pump (Figure 5).

This type of control unit enjoyed the advantages of being structurally simple, inexpensive, and highly adaptable to early-developed variable-rate spraying systems [60–62]. However, the disadvantages of slow response time, high fluctuation of pressure, and unstable spray performance limited its further applications. Because the pressure of this control unit was quadratically proportional to the flow rate, the pressure should be substantially increased or decreased to moderate the flow rate correspondingly. Generally, the increase in system pressure would lower the droplet size. The larger quantity of droplets and smaller size were beneficial to the coverage rate but inevitably increased the drift potential. On the

contrary, great pressure decreases were required to lower the flow rate. Meanwhile, the violent pressure fluctuations could dramatically change the droplet size, droplet velocity, and spray angle. GopalaPillai et al. [63] made a comparison between the pressure-based and the PWM-based flow rate control units. It was found that the flow rate measured at the 10% of duty cycle by the PWM-based unit could only be achieved through reducing the pressure from 207 kPa to below 6 kPa by the pressure-based one. Such a large pressure gap would substantially change the spray characteristics, thereby compromising spray efficacy. In order to ensure the performance of the sprayer with the pressure-based system, the flow rate variation from maximum to minimum should be limited within 25%. Anglund and Ayers [64] investigated the performance of a pressure-based variable-rate ground sprayer. Due to the delay in signal transmission and mechanical response, it was demonstrated that there was a lag time (approximately 2.35 s) between the sending out of the spray pressure control command and the actual pressure change response of the system.

To resolve the problems existing in this unit, many researchers attempted to modify it. The predetermined-time method was proposed to solve the system's delay phenomenon [64,65]. King and Wall [66] used the variable-frequency motor controller to adjust the frequency and effective magnitude of the voltage to control the rotation speed of the electric motor and complete the pressure and flow rate regulation.

### 3.1.2. PWM-Based Flow Rate Control Unit

The PWM-based flow rate control unit was the most widely used unit presently. Its principle was to adjust the duty cycle of the PWM signal to control the opening and closing time ratio of the solenoid valve to realize the variable output (Figure 5). The PWM duty cycle was the proportion of the valve open time to the whole cycle in a pulse period, which ranged from 0% to 100%. The 0% duty cycle meant the valve was fully closed while the 100% duty cycle indicated the valve was fully open.

Compared with the pressure-based flow rate control unit, the PWM-based unit ensured a relatively constant pressure while changing the flow rate, leading to consistent spray characteristics [8,61,67–69]. For example, Grella et al. [70] demonstrated that the on-off effect of the PWM system did not affect the uniformity of spray coverage within the canopy under real field conditions regardless of duty cycle and forward speed adopted. Moreover, it had a quick response time and high accuracy. However, droplet sizes were inconsistent at low duty cycles under low operating pressures [8,61,69–72]. Butts et al. [69] recommended that the PWM sprayer should only be equipped with non-venturi nozzles and operate at a pressure of not less than 276 kPa and a duty cycle of not less than 40% to ensure proper spray performance. At present, most control units were operating at 10 Hz frequency. GopalaPillai et al. [63] validated that at this frequency this control unit performed poorly with respect to spray uniformity along the travel direction when operating at a high travel speed and a low duty cycle. They suggested the optimum frequency should be established to coordinate well with a real-time sensing system. Jiang et al. [73] demonstrated that the higher the PWM frequency was, the better the uniformity of spray distribution would be. However, the higher the PWM frequency was, the shorter the life span of the solenoid valve would become.

### 3.1.3. Changing a Nozzle Orifice

Another novel variable flow rate means was achieved by changing the nozzle orifice's size and shape (Figure 5). This technology emerged within the past two decades. Firstly, Bui [74] and Womac [75] tested a variable-orifice nozzle and found it was possible to change the flow rate by varying the nozzle orifice's size. However, the droplet size spectra also changed accordingly. Later, an actual variable-rate nozzle was successfully designed and evaluated for the first time by Womac and Bui [76]. It changed the nozzle orifice's size using a diaphragm to move a metering plunger in and out of a sleeve when increasing or decreasing the liquid pressure (Figure 6). Bui [77] developed a similar nozzle that could modify its orifice's shape by moving a metering plunger in and out of a sleeve. These

two nozzles all depended on liquid pressure changes to alter the spring force acting on the metering plunger, so as to realize a variable flow rate, so they were still reactive in nature [78] and they had the same disadvantages as the pressure-based variable flow rate unit. It was shown that these nozzles could control the flow rate, but spray characteristics, such as spray deposition uniformity, droplet size distribution, and spray pattern angle, varied with the flow rate.

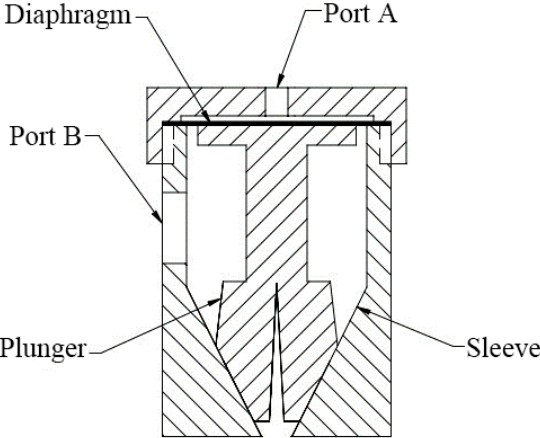

**Figure 6.** Schematic diagram of a variable nozzle. Port A: control liquid pressure, Port B: supply line pressure [76].

Some researchers noticed their shortcomings and attempted to modify them. For illustration, using air pressure instead of liquid pressure to act on the spring, Luck et al. [79] advanced the nozzle, which could move the metering plunger by changing the air pressure on the diaphragm while ensuring constant liquid pressure. Luck et al. [78] further evaluated its spray characteristics at constant pressures. Their test results showed a turndown ratio (2.4:1) for the flow rate, an acceptable coefficient of variation (CV <15%) for the spray pattern, and slightly higher droplet spectra than the manufacturer-specified value were achieved at each pressure in the range from 138 to 414 kPa. Overall, this nozzle could control the flow rate, but it needed further modifications to ensure spray pattern and droplet size consistencies.

*3.2. Variable Concentration Control Unit*

The above three control units needed to mix pesticide concentrates with a carrier (usually water) in the pesticide tank before spraying. However, this tank mixing method was problematic, due to the disposal of excessive leftovers of the tank mixture and the applicator's exposure to concentrated pesticides [80]. These problems could be solved with a variable concentration control unit (In-line injection system). Specifically, the pure water and pesticide concentrates were stored separately. During the spray application, the pesticide concentrates were injected or inhaled into the mixing chamber of the spray line or nozzle through the pesticide pump to mix with constant-rate-discharged water online. This unit could regulate the concentration of the pesticide mixture in real time by changing the flow rate of the pesticide pump.

The in-line injection system was developed and investigated over the past several decades [80–85]. However, its shortcomings included a long lag time, low mixture uniformity, inaccurate chemical concentration, and high application rate errors [83]. Zhang et al. [84] developed an experimental premixing in-line injection system, which made use of a metering pump and a water pump, respectively, to pump chemicals and pure water into a premixing tank through a static mixer first, and then transferred the mixture to the buffer tank (Figure 7). Zhang et al. [85] investigated its chemical concentration accuracy and spray mixture uniformity, finding that this system was proven to have the capability to

provide accurate and consistent uniformity of mixtures and have the potential to minimize tank mixture leftovers.

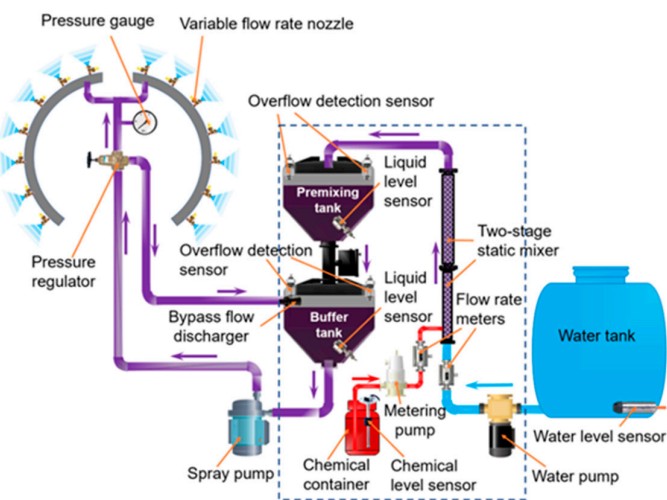

**Figure 7.** Schematic diagram of the premixing in-line injection system [84].

### 3.3. Variable Air Supply Control Unit

Real-time adjustment of air supply based on crop characteristics, plant diseases and pests and their severity, and meteorological conditions was another important research direction for air-assisted variable sprayers. The air-assisted sprayer could blow the leaves to roll, and improve the droplet penetration and the uniformity of pesticide deposition inside and outside the canopy [86]. The air supply should match well with the crop characteristics and meteorological conditions to ensure good spraying performance. However, at present, a large number of developed variable sprayers only realize variable flow rate applications at a constant air supply rate, which could easily cause droplets not to penetrate the canopy well or to drift when the air volume is insufficient or too large.

Air direction, air velocity, and air volume were three factors in variable air supply. At present, there are four methods for controlling the air supply (Table 3): adjusting the fan speed [87,88], adjusting the air inlet area, adjusting the air outlet area [89,90], and adjusting the distance between the air outlet and the canopy [91].

**Table 3.** Control methods and their relationships.

| Control Method | Adjusting Direction | Air velocity Change | Air Volume Change |
|:---:|:---:|:---:|:---:|
| adjust fan speed | ↑ [1] | ↑ [1] | ↑ [1] |
| adjust air inlet area | ↑ [1] | ↑ [1] | ↑ [1] |
| adjust air outlet area | ↑ [1] | ↓ [2] | ↑ [1] |
| adjust the distance | ↑ [1] | ↓ [2] | ↓ [2] |

[1] ↑indicates "increase". [2] ↓indicates "decrease".

Khot et al. [89] retrofitted an axial-fan air-blast sprayer with adjustable air-diverting louvres (Figure 8). Variable air supply was achieved by moving the air-diverting louvre plate to adjust the air outlet area. Figure 8 shows the air outlet's fully open state. This retrofitted sprayer provided a new method for air adjustment, but it did not achieve real-time adjustment of the air supply based on crop characteristics. Furthermore, when the air outlet area was reduced, the air speed would increase while the air volume was reduced. Therefore, it failed to achieve independent control of air speed and air volume [92]. Li et al. [88] developed a real-time variable-flow-rate variable-air-supply sprayer with 40 nozzles and eight fans. Each nozzle, coupled with a solenoid valve, independently achieved variable flow rate in real time by adjusting the duty cycle of PWM signals based on the canopy volume detected with the laser-scanning sensor. In addition, each brushless

fan independently achieved variable air volume by changing its voltage to adjust its speed according to the canopy volume. This independent control of the eight-fan speed method increased air volume while maintaining a constant air speed. Jiang et al. [90] designed a single-fan multi-duct bypass-air-regulation sprayer, which adjusted air volume and air velocity in real time by controlling the disc valve opening degree at the air outlet based on the crop volume detected with ultrasonic sensors. It was shown that this variable-air-volume sprayer immensely reduced drifts and ground losses of pesticides and improved canopy depositions. Miranda-Fuentes et al. [91] developed an air blast sprayer with two axial fans mounted on a tower-like structure with four mobile air outlets. This sprayer could adjust the air outlet position based on the canopy shape detected with ultrasonic sensors to keep the distance between the air outlet and the canopy constant. It was found that because the movement mechanism was composed of pulleys and a rack-and-pinion system, the actuator speed was low, resulting in poor coordination with the sprayer's forward speed.

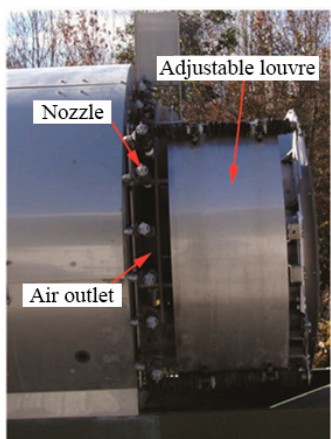

**Figure 8.** Variable air supply by adjusting air outlet area [89].

### 3.4. Variable Nozzle Position and Type

The distance between the nozzle and the target was one of the important technical parameters affecting spray performance. According to the target structural parameters or meteorological conditions, the real-time adjustment of the operating parameters of the sprayer, such as nozzle position and nozzle type, was one of the objectives of the intelligent variable sprayer. Osterman et al. [93] developed a variable-nozzle-position air-assisted orchard sprayer, which could adjust three hydraulically movable spraying arms based on real-time detection parameters of the target structure from a laser scanner to better cover the target tree (Figure 9). This sprayer could immensely reduce pesticide drift and ground deposits, and improve application efficiency. Miranda-Fuentes et al. [91] developed a new air-assisted sprayer that could adjust nozzle positions based on the olive characteristics to reduce spray drift and off-target application. Balsari et al. [15] developed an intelligent sprayer which was able to automatically select the suitable nozzle type (conventional nozzle and air induction nozzle) to spray according to the natural wind conditions at the spraying time (Figure 10). A currently-developed variable-nozzle-position intelligent sprayer achieved nozzle up–down and left–right movement and nozzle angle adjustment relying on the mechanical structure driven by either the electric [94] or hydraulic system [93]. The time required to complete these mechanical actions was relatively long, which did not match the travel speed of the sprayer. Consequently, it was difficult to achieve real-time adjustment of the nozzle position in spray application [95].

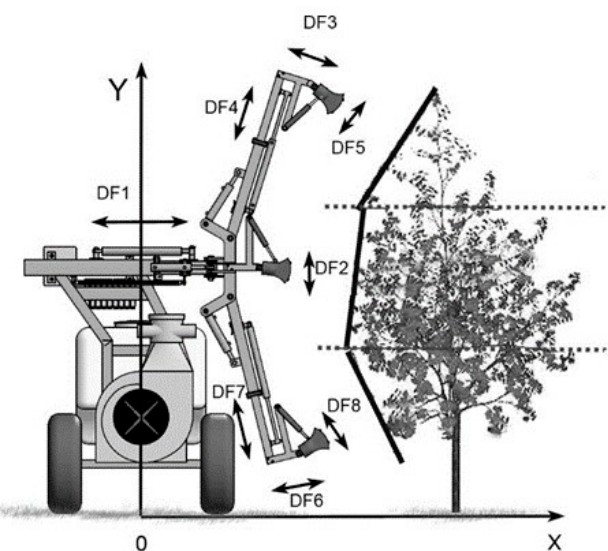

**Figure 9.** Variable nozzle position with eight degrees of freedom [93].

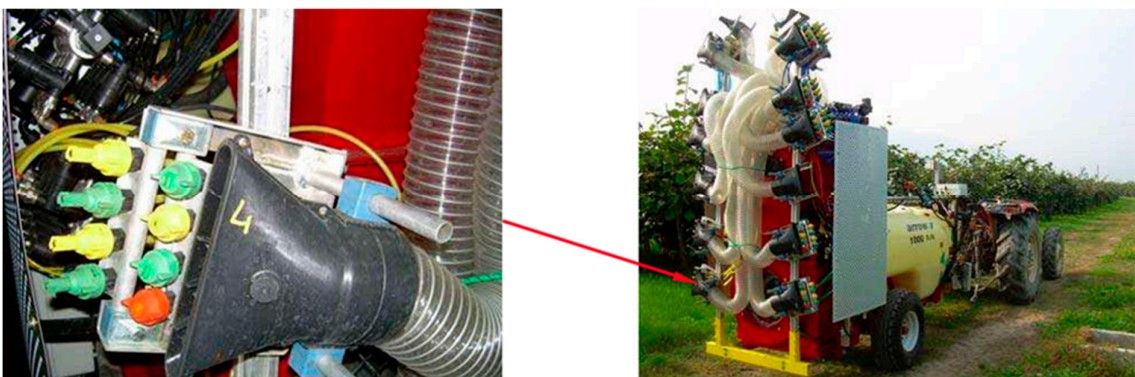

**Figure 10.** Variable-nozzle-type sprayer prototype [15].

Among the above-mentioned three variable flow rate control units, the PWM-based unit was the most widely used one, followed by the pressure-based unit. The variable concentration control unit was preliminarily verified in the laboratory. The variable air supply control unit was tested in both laboratory and field experiments, but is still far from practical application. In terms of the four methods of changing the air supply, adjusting the fan speed and adjusting the air outlet area were the most commonly used.

## 4. Signal-Processing Algorithm

In the process of variable-rate application, detection of target parameters, data processing, decision making, and variable output control happened almost simultaneously. In developing a variable-rate sprayer, it was essential for its detection unit to be capable of accurately sensing crop parameters and for its variable control unit to be capable of adjusting the operating parameters in real time. However, it was more important to establish a high-speed algorithm for target parameter post-processing and variable-output decision making. The algorithm served as a bridge between the detection unit and the variable control unit. Its main functions were to collect and save data from detection units, to filter and process data, and to make decisions for variable control units.

### 4.1. Key Points in Establishing an Algorithm

Ensuring the high efficiency of an algorithm was one fundamental requirement for developing a real-time variable sprayer because the interval between the time when the crop was detected and the time when the nozzle began to spray was extremely short. The

length of this interval counted on the travel speed of a sprayer and the distance from the detection sensor to the nozzle. The algorithm should ensure that its post-processing of data and decision-making of variable outputs could be completed quickly and accurately in such a short time. Nonetheless, there were many factors affecting the operating speed of an algorithm, for example, the quantity of the data to be processed and the establishing platform of an algorithm. For instance, Rovira-Más et al. [96] set up an algorithm to process data from stereovision. The 3D data were of such a magnitude that the algorithm could not handle and save data in real time. Although the algorithm was modified by the method of processing one image at one time and then deleting this image after extracting the meaningful information, it was still inefficient. Chen et al. [24] formulated an algorithm with an establishing platform named LabView that had a large internal structure, resulting in a low efficiency of the algorithm.

Besides the efficiency of an algorithm, obtaining the correct raw data was also crucial for establishing an algorithm. The unwanted data might come from the ground, sky, or objects beyond the target area. Some erroneous data, possibly resulting from electromagnetic sources, mechanical vibrations, and moving leaves, had to be filtered by means of comparison with preset thresholds and selection of appropriate filters [97].

### 4.2. Application Dosage Models

The lack of a dominant model to calculate the optimal application dosage became a big obstacle to establishing an algorithm. The most common recommendation of the application rate was the ground area (GA)-based model usually appearing on pesticide product labels as a constant application dose. This dose expression was only related to the ground area occupied by the target but not to the individual differences in the geometric characteristics of the target canopy. As a result, it was not appropriate for variable spraying.

For variable spraying, different dose rate expressions were proposed. Byers et al. [98] first proposed the concept of Tree Row Volume (TRV) which assumed that a tree row was a cuboid whose volume could be used to calculate the total canopy volume per unit ground area. The TRV dose rate was the application rate per unit canopy volume. Thus, it was notable that this dose rate considered the canopy height and width. Later on, to obtain consistent deposits within the canopy with different canopy densities, Sutton and Unrath [99] modified this model by adding a canopy density factor ranging from 0.7 (open canopy) to 1 (extremely dense canopy) into the TRV calculation equation. Furness et al. [100] simplified TRV with a unit canopy row (UCR) defined as 1 m high × 1 m wide × 100 m row length. The UCR model did not need to provide the required ground area and row spacing as parameters in the TRV calculation, which was helpful for its promotion. It was suggested that the TRV application rate significantly reduced pesticide use while achieving comparable deposition and coverage on leaves, in comparison to the GA application rate [14,28]. Based on the TRV model, Garcerá et al. [101] developed a support tool named CitrusVol to help citrus growers choose the appropriate application rate in their orchards. This tool took into account the canopy size, the foliar density, the pest/disease type, and the PPP to be applied. It was shown that this tool improved the application efficiency without affecting the biological efficacy.

Another dose rate expression was the leaf-wall-area (LWA) model which regarded the crop as a vertical wall facing the nozzle whose area could be used to calculate the total wall area per unit ground area (Figure 11). This model was related to the canopy height and row spacing and accepted by agrochemical manufacturers, crop growers, and European regulatory authorities. European agrochemical manufacturers agreed to use the LWA dose rate for new pesticide registration. European regulatory authorities attempted to use this rate on product labels [102]. Later on, EPPO [103] selected the LWA method as the most accurate way to determine the optimal application rate. Walklate and Cross [104] conducted deposit trials in UK pome-fruit orchards to examine the LWA dose rate, concluding that the LWA dose rate was efficacious and safe to use. Based on the TWA model, Gil et al. [105] developed a tool named DOSAVINA to calculate the optimal volume rate. This method

took the effects of leaf density and canopy width into consideration. It was noted that a variable-rate sprayer based on this method could reduce pesticide use by more than 20% while still achieving satisfactory spraying performance.

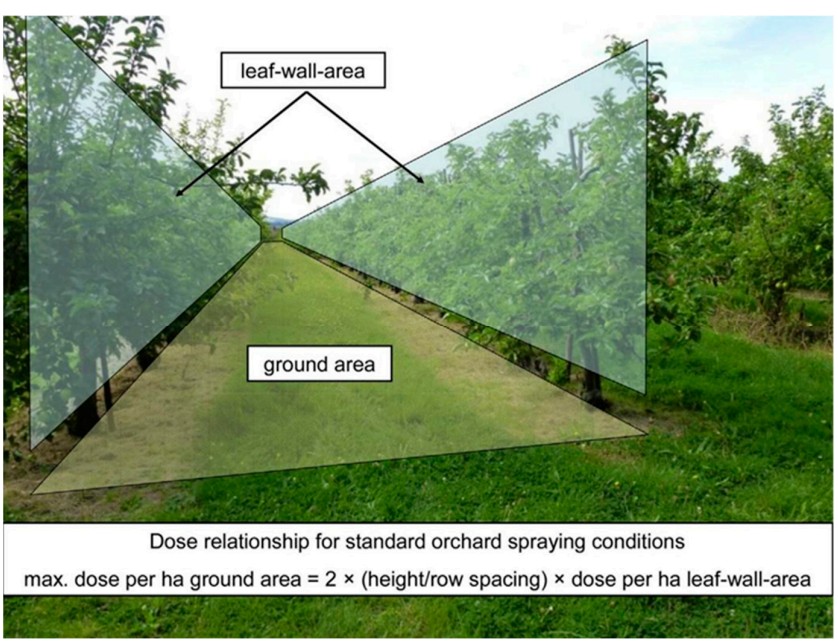

**Figure 11.** Relationship between LWA and GA [104].

A third dose rate expression was the canopy-height (CH) model simplified from the LWA model in standard orchards where the row spacing was constant [106]. This model was solely related to the canopy height. However, in actual orchards, the parameters (canopy width, row spacing, canopy density, leaf area index, and pruning agronomy) were not constant, making this model less popular.

*4.3. Application Mode*

Generally, there were three application modes: the ON/OFF application mode, the discrete application mode, and the continuous application mode. To some extent, the application mode represented the development level of the variable-rate sprayer.

The ON/OFF application mode was largely employed in the early development of variable-rate sprayers. It realized intermittent spray by setting an output threshold of detection sensors and independently controlling the on–off action of each solenoid valve (Figure 12b). Giles et al. [107] developed an orchard air-assisted atomizer with ultrasonic range transducers and on–off solenoid valves. The atomizer could interrupt the spray when no tree was detected. The field tests showed that the sprayer could reduce spray solution use by 28–52% at different stages of leaf development. However, this mode did not actually achieve variable spraying because the flow rate remained constant when spraying.

The discrete application mode could not only judge whether the nozzle needed to spray according to the presence or absence of the canopy but also apply different dose rates according to the size of the canopy detected (Figure 12c). This mode was a transition to the continuous application mode. Moltó et al. [108] developed an automatic sprayer with two ultrasound sensors and several electro-hydraulic valves, able to apply a high dose, a low dose, or no dose based on the leaf mass. It was discovered that this sprayer reduced pesticide use by 37% without affecting the treatment efficacy.

The continuous application mode adjusted the flow rate in a continuous (non-discrete) manner (Figure 12d). Solanelles et al. [109] developed a sprayer using ultrasonic sensors for the canopy width estimation and proportional solenoid valves for the flow rate adjustment. This sprayer proportionally adjusted the flow rate based on the ratio of actual canopy width

detected from sensors to the maximum canopy width corresponding to the full dose rate. Chen et al. [24] developed a variable-rate sprayer consisting of a laser-scanning sensor and PWM solenoid valves. This sprayer regulated the flow rate in a continuous manner by adjusting the duty cycles of the PWM solenoid valves based on the variation in canopy characteristics measured by the sensor. In order to achieve the accurate application of pesticides, Berk et al. [110] developed a real-time fuzzy logic system to linearly change the flow rate from 0% to 100% through the proportional solenoid valves according to the detected signals from the sensor.

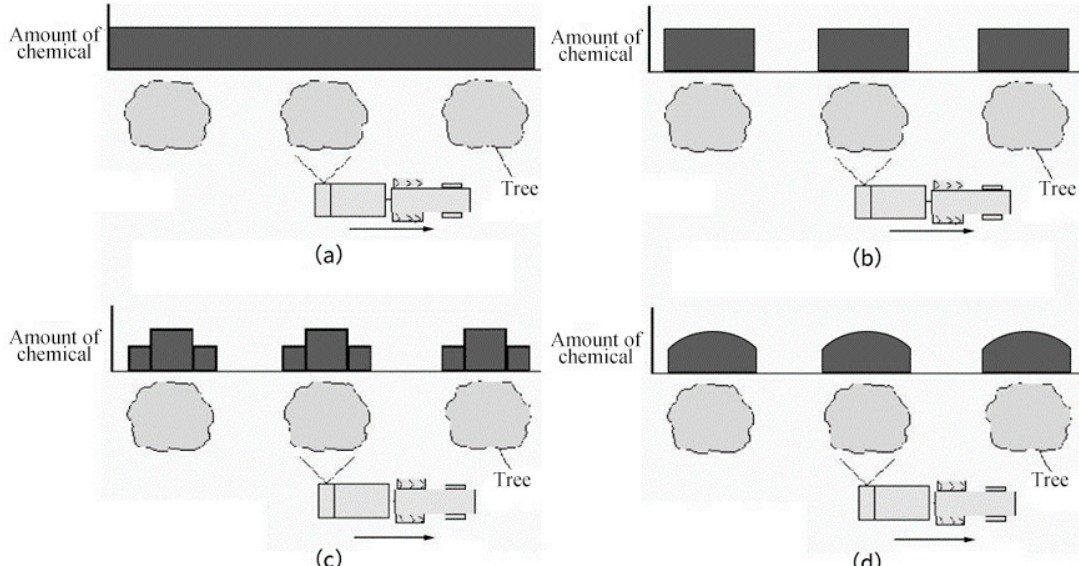

**Figure 12.** Quantity of chemicals applied by a constant-rate sprayer (**a**) and variable-rate sprayers with the ON/OFF application mode (**b**), the discrete application mode (**c**), and the continuous application mode (**d**) (partially cited from Moltó et al. [108]).

In conclusion, the TRV and LWA dosage models were accepted by national regulatory authorities and agrochemical manufacturers, and widely used. The continuous application mode adjusted the flow rate in a continuous way, realizing the variable-rate application in a true sense.

## 5. Research Status on an Orchard Variable-Rate Sprayer

Multiple variable-rate sprayer prototypes were developed using different detection sensors, variable control units, and signal processing algorithms. Among these, the variable-rate sprayer with ultrasonic sensors or laser scanning sensors, PWM-based flow control units, as well as TRV-based or LWA-based algorithms were the most promising options.

Although the ultrasonic sensor had shortcomings, it was typically used to detect the target occurrence and measure target geometrical parameters in variable-rate spray systems. Giles et al. [107] developed an orchard air-assisted atomizer with ultrasonic range transducers and on–off solenoid valves, with these transducers used to detect the presence and measure the height and width of targets and with these valves used to interrupt the spray when no tree was detected. The field tests showed that the sprayer could measure the tree width and height with smaller than 10% average error and reduce spray solution use by 28–52% at different stages of leaf development. Afterwards, many researchers used ultrasonic sensors to retrofit conventional sprayers and evaluated their performance [14,21,28,109,111]. Jeon and Zhu [111] developed an intelligent real-time variable-rate spray system with two vertical booms and ultrasonic sensors to tune spray discharge automatically on the basis of the liner canopy size (Figure 13). It was indicated by field test results that compared with the constant rate and TRV-based-rate spray methods, this sprayer saved liquid use by up to 86.4% and 70.8%, respectively [111].

Maghsoudi et al. [112] developed a variable-rate sprayer using three ultrasonic sensors to detect the variation in canopy structure and a multilayer perceptron (MLP) network algorithm for the canopy volume estimation. This sprayer could adjust the valve opening based on the canopy volume in real time to achieve a variable-rate discharge. It was demonstrated by field tests that compared with the traditional constant-rate sprayer, this sprayer achieved 41.3%, 25.6%, and 36.5% spray savings at the top, middle, and bottom of the tree, respectively.

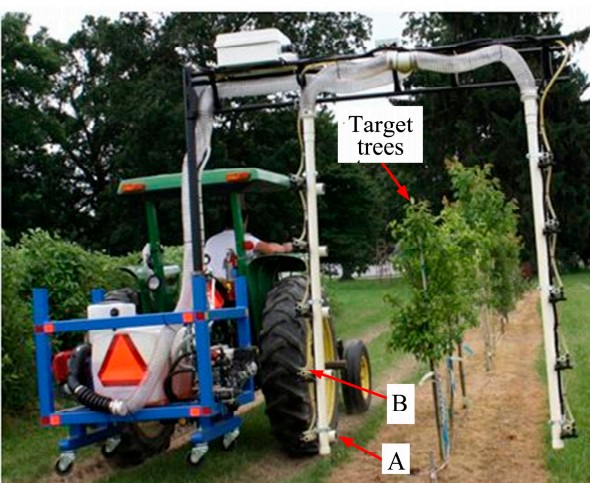

**Figure 13.** The real-time variable-rate sprayer and its components (**A**—ultrasonic sensor, **B**—spray nozzle with PWM solenoid valve) [111].

Since 2012, a research team led by Professor Heping Zhu has been devoted to developing a variable-rate precision spray system with a laser-scanning sensor and a PWM-based digital flow control unit (Figure 14) [1,24,86,113–116]. Liu et al. [113] developed a PWM-based multi-nozzle independent control digital flow unit for this intelligent sprayer. The laboratory tests verified that this unit could precisely produce the required PWM signal and control linear spray outputs. To validate this sprayer's performance, Shen et al. [1] conducted a field experiment that demonstrated that this sprayer operating in the variable-rate mode (VRM) reduced the spray volume by 12.1% to 43.3% while achieving 30% to 55% larger coverage areas per quantity of spray deposits, compared with that operating in the constant-rate mode (CRM). To promote its successful large-scale adoption by growers, sprayer manufacturers, pesticide manufacturers, and regulating agencies, Zhu et al. [115] conducted field evaluations in three commercial nurseries from pest control and economic feasibility perspectives, finding that compared with the constant-rate sprayer, this sprayer saved liquid use by 29.7% to 77.6% while still achieving satisfactory insect pest control outcomes. Chen et al. [116] further validated that this intelligent sprayer could reduce pesticide use and avoid killing natural enemies of pests, thereby realizing more effective control of insects and diseases and protection of the environment and ecology. Salcedo et al. [86] also validated its performance in a two-year apple orchard, noting that this system could provide adequate foliar deposition and coverage on trees while saving pesticides.

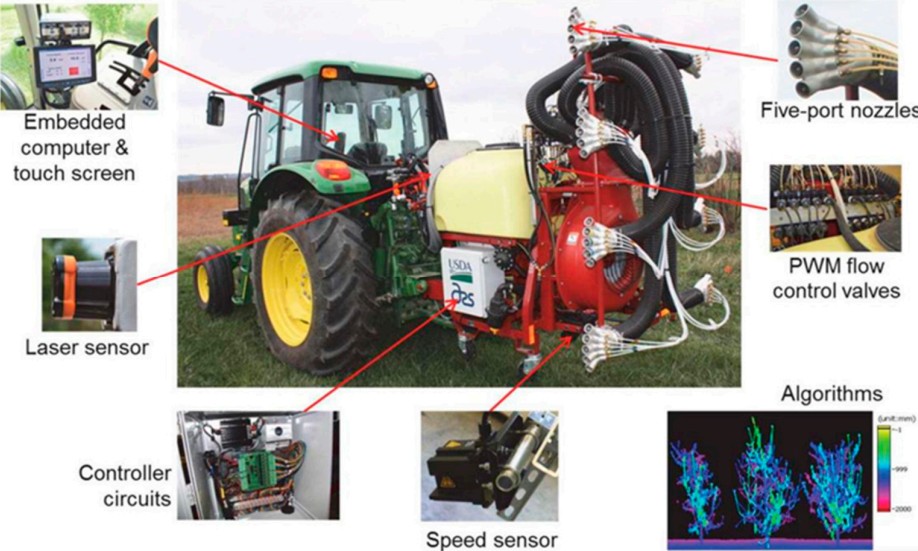

**Figure 14.** The laser-based variable-rate spraying system [115].

As in Europe, public institutions and sprayer manufacturers were joint forces to develop and promote variable-rate spraying systems among farmers. An example could be observed in the OPTIMA project (http://optima-h2020.eu), integrated within a European strategy to reduce the consumption of PPPs [117]. A part of this international project focused on developing efficient spraying technologies in apple tree treatments [118]. In this way, the Technical University of Catalonia (UPC, Spain) and Fede Sprayers (Pulverizadores Fede SL, Spain) worked together on the design of an intelligent orchard sprayer. This sprayer included three ultrasonic sensors per side, connecting each one to a particular section of the boom, as well as a GPS system and a speed sensor. During a spray pass application, each sensor sent data to a computer to determine the corresponding optimal volume rate to spray in each section in real time. Calculations were performed using the Tree Row Volume (TRV) method with additional information regarding the leaf density. There was one solenoid valve in each section controlling the outgoing flow rate. When the computer estimated the rate application, it sent a signal to every valve to regulate the final flow rate. The field experiments showed that this system could save a minimum of 23% of the quantity of liquid while maintaining biological efficacy compared to a conventional treatment (Figure 15).

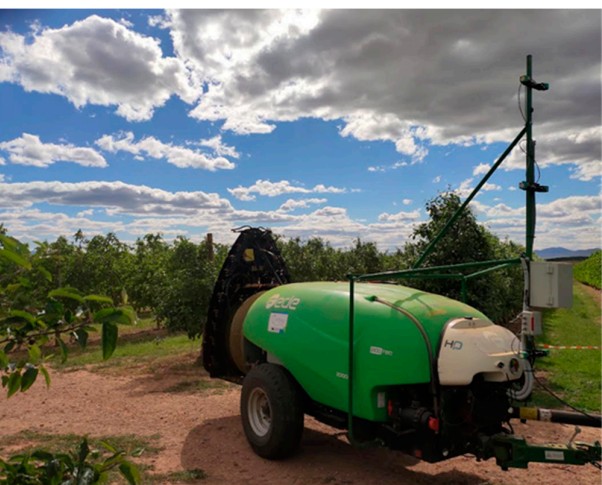

**Figure 15.** Prototype of the variable-rate sprayer developed in the European OPTIMA project.

**6. Further Prospects and Conclusions**

*6.1. Further Prospects*

Future prospects are as follows.

(1)    Accurate real-time detection of canopy density is a future research direction. Although researchers have verified the relationship between the intensity of ultrasonic echo and the foliage density in the laboratory, whether it can be used for online detection still needs in-depth research. Further verification of the internal structure perception and the canopy density estimation method based on laser-scanning sensors are also needed.

(2)    Accurate real-time detection of plant diseases and pests and their severity is another research direction. Optimizing the image technology based on machine vision, improving the efficiency of image segmentation and classification algorithms, and realizing the recognition of plant diseases and pests and the real-time variable application of pesticides are future research focuses. In addition, researchers need to pay due attention to deeply studying the electronic nose technology, reducing the cost of an electronic nose, and improving its detection accuracy and efficiency.

(3)    Real-time acquisition of meteorological conditions and real-time adjustment of the operating parameters of variable sprayers based on these conditions will be another research direction regarding intelligent sprayers.

(4)    It is worth modifying and optimizing the air demand theory, developing the air regulation device with independent control of air speed and air volume, and improving the efficiency of the air regulation device.

(5)    In terms of the variable concentration control unit, chemical concentration accuracy and spray mixture uniformity demand further laboratory and field validation.

*6.2. Conclusions*

Variable-rate sprayers have been developed for more than three decades since Giles et al. [107] developed an orchard intermittent variable-flow-rate air-assisted sprayer with ultrasonic range transducers and on–off solenoid valves. The research status of the variable sprayer is summarized below:

(1)    In terms of target parameter detection, ultrasonic sensors and laser-scanning sensors are widely used for target position and canopy volume detection. Variable-rate sprayers based on these two sensors have been tested in the laboratory and field and are being commercially applied. The detection of canopy density based on these two sensors has been preliminarily verified in laboratory and field experiments, while the detection of plant diseases and insect pests and their severity based on machine vision and the electronic nose is still in the stage of laboratory research. Besides, the variable-rate sprayer based on meteorological conditions is in the "conception" stage.

(2)    Among the three variable flow rate control units, the PWM-based unit is the most widely used one, followed by the pressure-based unit. The variable concentration control unit has been preliminarily verified in the laboratory. The variable air supply control unit has been tested in laboratory and field experiments. However, it is still far from practical application. Among the four methods of changing the air supply, adjusting fan speed and adjusting air outlet area are the most commonly used options.

(3)    The TRV and LWA dosage models have been accepted by national regulatory authorities and agrochemical manufacturers, and are widely used. The continuous application mode adjusts the flow rate in a continuous way, which realizes the variable-rate application in a true sense.

A laser-based variable-rate intelligent sprayer equipped with PWM solenoid valves to tune spray outputs in real time based on target structures has the potential to be successfully adopted by growers on a large scale in the foreseeable future. It will be a future research direction to develop a smart multi-sensor-fusion variable-rate sprayer based on target crop

characteristics, plant diseases and pests and their severity, and meteorological conditions while achieving multi-variable control.

**Author Contributions:** Conceptualization, Z.W. and Y.S.; methodology, Q.L. and Y.Z.; investigation, Q.D.; resources, J.S.; writing—original draft preparation, Z.W.; writing—review and editing, Z.W.; visualization, Q.H.; supervision, X.X., R.S., Z.Z. and E.G.; project administration, X.X.; funding acquisition, X.X., Z.W. and Z.Z. All authors have read and agreed to the published version of the manuscript.

**Funding:** This research was funded by the Agricultural Science and Technology Innovation Project of the Chinese Academy of Agricultural Sciences, Crop Protection Machinery Team (grant No. CAAS-ASTIP-CPMT), the Agricultural Science and Technology Innovation Project of the Chinese Academy of Agricultural Sciences (grant No. CAAS-XTCX2018023), Jiangsu Province and Education Ministry Co-sponsored Synergistic Innovation Center of Modern Agricultural Equipment Project (grant No. XTCX1004), Key Research and Development Project of Shandong Province (grant No. 2022SFGC0204), and the Think Tank Youth Talent Program of China (Grant No. 2022ZZ041876).

**Institutional Review Board Statement:** Not applicable.

**Informed Consent Statement:** Not applicable.

**Data Availability Statement:** Not applicable.

**Acknowledgments:** Thanks to Yu Qingxu and Zhang Guangyue for their assistance in the literature search. Thanks to the editors and experts for suggesting changes to this article.

**Conflicts of Interest:** The authors declare no conflict of interest.

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
