# Peer review of "Key Technologies for an Orchard Variable-Rate Sprayer: Current Status and Future Prospects"

_agronomy, doi:10.3390/agronomy13010059_

Round 1

Reviewer 1 Report

Thank you for your research. The subjects dealt with are very interesting and current in precision agriculture.

I would have expected a more schematic organization of the paper for a survey.

In addition, I would invite you to make some adjustments:

-        I would try to modify the title because it is not representative enough of the work;

-        The introduction is too short and the objectives and our contribution to the paper are not clearly exposed;

-        All references must be unified, always insert authors or not;

-        Table 1 must be verified, there are errors in the bulled lists;

-        The paper organization could be inserted at the end of the Introduction;

-        The variable-rate spray system is composed of: i) a detection unit, ii)a data processing algorithm, and iii) a variable control unit (as presented in the Introduction), but the structure of the paper does not respect this order;

-        All inserted figures from other papers are necessary?

-        For example, Figure 2 is taken from which paper? [15,22]?

-        The “Summary” for Sections 2, 3, and 4 is necessary? If you leave this section, I would try to deepen;

-        Figure 5 is yours? Or is it taken from another paper, which one?

-        In Section 3.1.3, why don’t you talk about anti-drift nozzles?  

-        Unify acronyms, for example, line 512 -> Ground Area (GA);

-        It should be clearer Section 5, with the same title as the paper.

-        I’d join a final section combing further prospects and conclusions. 

Reviewer 2 Report

The reviewer's remarks are found in the original pdf file

Reviewer 3 Report

The proposed review is able to resume the main variable-rate technologies available and studied for more than 30 years. The paper is well written and the topics are analyzed and resume in an easy but complete flow. The introduction section presents the topic and lists the technologies that will be discussed in the paper.

Here is a list of minor changings suggested:

LINES 14-31: I suggest removing the acronyms from the abstract section and using them only in the right sections of the paper

LINES 20-23: “Among the three variable flow rate control units, the pulse width modulation (PWM) based unit is the most widely used one, followed by the pressure-based unit. The variable concentration control unit is preliminarily verified in the laboratory.” - Please modify as: “Among the three variable flow rate control units, pulse width modulation resulted as the most widely used, followed by the pressure-based and by variable concentration which is preliminarily verified in the laboratory.”

LINE 80: “…laser scanning sensor…” I understand that you are referring to LiDAR. If I’m right, please use the same “definition” used in LINE 79 “…light detection and ranging…” or vice versa, or use the name “LiDAR” to be clearer, or in addition use the definition written in LINE 138: “…laser scanning sensor, one kind of LiDAR sensor”

LINES 82-83: “…was still an attractive choice.”. I suggest explaining why could result in an attractive choice here or please avoid it

LINES 85-87: I suggest removing Table 1 and adding in the text all the contents. In addition, please avoid acronyms (i.e. TOF used for LiDAR sensor, dot 1) in tables. In fact, for example, LINES 108-109 already presented the advantages of Ultrasonic sensors.

Figures: I generally suggest enlarging the text in all figures. As it is now it results are too small to be properly read.

LINE 116: “…large deviation of the volume estimation.” please add a value

LINE 128: “…TOF principle.” Please write here the full name of the acronym and not in Table 1

LINE 225: “…Botrytis cinerea…” please write the name in italic as “…Botrytis cinerea…”

LINE 295: “…out that Figure 5 was…” I think that the verb to be should be written in present form becoming “…out that Figure 5 is…”

LINES 331-351: in Paragraph 3.1.2 I suggest to add also a couple of lines to list the results obtained in Grella et al 2022 in which a 20 Hz PWM was tested for its spray coverage performances by evaluating different Duty Cycles and forward speeds

REF: Grella, M., Gioelli, F., Marucco, P. et al. Field assessment of a pulse width modulation (PWM) spray system applying different spray volumes: duty cycle and forward speed effects on vines spray coverage. Precision Agric 23, 219–252 (2022). https://doi.org/10.1007/s11119-021-09835-6

LINE 387: “…water…” please add “pure” becoming “…pure water…”

LINE 396: same comment as LINE 387
